# Human-Biting *Ixodes scapularis* Submissions to a Crowd-Funded Tick Testing Program Correlate with the Incidence of Rare Tick-Borne Disease: A Seven-Year Retrospective Study of Anaplasmosis and Babesiosis in Massachusetts

**DOI:** 10.3390/microorganisms11061418

**Published:** 2023-05-27

**Authors:** Eric L. Siegel, Nathalie Lavoie, Guang Xu, Catherine M. Brown, Michel Ledizet, Stephen M. Rich

**Affiliations:** 1Laboratory of Medical Zoology, Department of Microbiology, University of Massachusetts, Amherst, MA 01003, USA; esiegel@umass.edu (E.L.S.); gxu@umass.edu (G.X.); 2Graduate School of Biomedical Sciences, Tufts University, Boston, MA 02111, USA; nathalie.lavoie@tufts.edu; 3Massachusetts Department of Public Health, Boston, MA 02108, USA; catherine.brown@state.ma.us; 4L2 Diagnostics, LLC, New Haven, CT 06511, USA; mledizet@l2dx.com

**Keywords:** anaplasmosis, babesiosis, passive surveillance, tick-borne diseases

## Abstract

Tick-borne zoonoses pose a serious burden to global public health. To understand the distribution and determinants of these diseases, the many entangled environment–vector–host interactions which influence risk must be considered. Previous studies have evaluated how passive tick testing surveillance measures connect with the incidence of human Lyme disease. The present study sought to extend this to babesiosis and anaplasmosis, two rare tick-borne diseases. Human cases reported to the Massachusetts Department of Health and submissions to TickReport tick testing services between 2015 and 2021 were retrospectively analyzed. Moderate-to-strong town-level correlations using Spearman’s Rho (ρ) were established between *Ixodes scapularis* submissions (total, infected, adult, and nymphal) and human disease. Aggregated ρ values ranged from 0.708 to 0.830 for anaplasmosis and 0.552 to 0.684 for babesiosis. Point observations maintained similar patterns but were slightly weaker, with mild year-to-year variation. The seasonality of tick submissions and demographics of bite victims also correlated well with reported disease. Future studies should assess how this information may best complement human disease reporting and entomological surveys as proxies for Lyme disease incidence in intervention studies, and how it may be used to better understand the dynamics of human–tick encounters.

## 1. Introduction

The blacklegged tick (*Ixodes scapularis* Say) is the hematophagous arthropod vector of several described human pathogens. The most prevalent among these is the Lyme disease-causing spirochete *Borrelia burgdorferi* sensu stricto, which is responsible for 476,000 annual infections in the United States [1,2]. Babesiosis and anaplasmosis are two slightly rarer diseases transmitted by *I. scapularis* that have become nationally notifiable due to rising incidence and spatial coverage [3,4,5]. Babesiosis is a malarial-like disease caused by intraerythrocytic protozoa of the *Babesia* genus [6]. Disease in the New England region is attributed to *Babesia microti* (Piroplasmida: Babesiidae), and clinical manifestations range in severity from asymptomatic to fatal, with one-third of reported cases warranting hospitalization [7]. Anaplasmosis, caused by *Anaplasma phagocytophilum* (Rickettsiales: Anaplasmataceae), results in acute febrile disease, hospitalizing one-fourth of reported cases [8,9]. Coinfection with *B. burgdorferi* and other pathogens may worsen clinical outcomes of both diseases, and although cases may be life-threatening in the immunocompromised and elderly, most are subclinical, self-limiting, or manifest only with non-specific symptoms [10,11]. Consequently, diagnosis is complicated, and surveillance efforts focusing on human case reporting are affected by misclassification, loss to follow-up, and an underestimation of true burden [12,13].

Public funding is limited for interventions aimed at mitigating the tick-borne disease burden, so efforts must be targeted with surveillance evidence [14]. Accurately describing the spatiotemporal distribution of disease and exposure risk requires more than a strict reliance on epidemiological surveys. The intricate relationships between the pathogen, environment, vector, and human host must be investigated for a complete picture of risk. This is due to the challenges of accurately diagnosing and reporting human disease, the complexity of the pathogen enzootic cycle, and the sensitivity of ticks and their pathogens to ecological change [15,16,17]. To broaden the scope of investigation, human disease statistics are traditionally supplemented with active entomological surveillance, measuring density and pathogen prevalence in field-collected ticks, sampled with flagging/dragging methods or with the capture of small mammals in less populous areas where human–tick interactions are uncommon [18,19].

Passive surveillance systems have been implemented to bridge the gap between human case data and entomological surveys [20,21,22,23,24,25,26,27,28,29,30,31,32,33,34]. Seroprevalence studies in companion/wildlife species and citizen science have proven useful for studying risk determinants, promoting community engagement, and encouraging cross-sector collaboration by creating partnerships between veterinarians, physicians, and medical entomologists [20,21,22]. Pathogen testing of human-biting ticks is a method that has been shown to be particularly predictive of the incidence of human Lyme disease [26]. It can provide large amounts of telling data as a resource-friendly, early warning indicator in emergent regions and a risk-mapping tool in endemic regions [31,32,33,34]. Tick testing documents the key aspects of interactions between ticks and people that mediate disease acquisition, including tick infection status, attachment time, and the geographical location of the bite (whereas human disease statistics are solely tied to residence) [33]. Despite the value of the individual and larger-scale information that can come from these systems, as proxies for disease incidence in intervention studies or tools to better understand human-tick encounters, the centers for disease control and prevention (CDC) and leaders among human physicians do not recommend that tick-bite victims utilize these services. This is largely to prevent overtreatment and ensure that patients appropriately monitor for symptoms of disease following exposure [7,35]. Further, the correlation made between Lyme disease incidence and tick testing measures has yet to be extended to other diseases.

TickReport was developed in 2006 at the Laboratory of Medical Zoology at the University of Massachusetts, Amherst to provide affordable, timely information to tick bite victims and public health agencies regarding tick–human encounters. Since 2020, the TickReport testing service has been provided by MedZu, Inc. under a licensing agreement with the University of Massachusetts. Testing results, for both individual ticks and aggregated statistics, are made available in real-time and are easily accessible online [36]. The present study retrospectively analyzed the last seven years of human biting *I. scapularis* submissions to TickReport alongside human babesiosis and anaplasmosis incidence as reported by the Massachusetts Department of Health. Town-level correlations were made across space and time to (1) evaluate whether the previously made associations between human-biting tick submissions and human Lyme disease extend to less prevalent, endemic diseases and (2) further ascertain the role of passive tick testing within tick-borne disease surveillance systems.

## 2. Materials and Methods

### 2.1. Institutional Review Board Statement

This project (1330108-1) was submitted to the Massachusetts Department of Public Health Institutional Review Board, and was determined not to meet the definition of human subject research on 12 October 2022.

### 2.2. Tick Testing Data

Tick samples, whole or partial, were received in plastic bags or vials for species identification and pathogen testing at the laboratory in Amherst, MA (Figure 1). Individuals submitting ticks volunteered information including removal date, towns of residence and bite, host information (species, age, sex), and if/where/for how long the tick was attached. Only human-biting *I. scapularis* were considered. To account for travel, correlations were restricted to submissions with matching towns of bite and residence [26]. Species identification was based on published identification keys [37,38,39]. *I. scapularis* was differentiated from other species of the *Ixodes* genus with a species-specific TaqMan real-time polymerase chain reaction (PCR) assay. Infection status with *B. microti* and *A. phagocytophilum* was assessed with a multiplex TaqMan PCR assay targeting different genes (Appendix A) [29]. Pathogen prevalence of nymphs and adults was evaluated over space at the county level and calculated by year and in total, with 95% confidence intervals [40]. The characteristics of attachment were also described, including where on the body bites were reported, attachment times, and victim ages. Self-reported attachment times, reported in 60 min increments, were compared between tick life stages with the Kruskal–Wallis H test, and laboratory-assessed engorgement proportions were compared with a chi-squared test for the difference in proportions.

### 2.3. Human Disease Data

Human case data were obtained from the Massachusetts Department of Health. Cases were aggregated at the town level and included the number of anaplasmosis and babesiosis cases for each year 2015 through 2021. Counts were classified as “confirmed or probable” as defined under the CDC National Notifiable Diseases Surveillance System (NNDSS) [41,42]. Confirmed and probable cases included clinical evidence. Probable cases met the criteria for clinical presentation and had supportive laboratory results, and confirmed cases had laboratory-confirming diagnostics. A subset of “suspected” cases was also included. These had laboratory evidence of current infection at the time of a laboratory report without any clinical information.

Laboratory evidence of human babesiosis and anaplasmosis followed guidelines outlined in the NNDSS: babesiosis supportive diagnostics followed one of four options: *B. microti* Indirect Fluorescent Antibody (IFA) immunoglobulin G (IgG) antibody titers; a positive immunoblot IgG; *B. divergens* IFA total Ig or IgG antibody titer; or *B. duncani* IFA total Ig or IgG antibody. Babesiosis-confirming diagnostics required one of the following: detection of *B. microti* in whole blood PCR; isolation of *Babesia* from whole blood by animal inoculation; detection of *Babesia* spp. via nucleic acid sequencing; or microscopic visualization [41]. Supportive laboratory evidence of anaplasmosis followed evidence of elevated antibodies reactive with *A. phagocytophilum* antigens by IFA/other serological evidence or microscopic visualization of morulae. Confirming diagnostics required more stringent IFA results; detection of *A. phagocytophilum* by PCR; antigen presence in biopsy or autopsy samples; or isolation in cell culture [42].

The provided counts were grouped. Confirmed and probable were separate from suspected cases. The total of both groups was used to correlate tick measures with disease, but looked at amongst themselves in an analysis of reporting dynamics. The ratio of suspected to confirmed and probable case rates was used to compare reporting across space, and Spearman’s Rho (ρ) was used to evaluate case reporting dynamics across time.

### 2.4. Spatial Mapping

Tick submission and human disease rates per 100,000 population were mapped using a shapefile obtained from MassGIS (Boston, MA, USA) [43]. For rate calculations, population was derived from the UMass Amherst Donahue Institute Population Estimates Program. Aggregated rates were calculated with the mean population for each town 2015–2021 [44]. Global Moran’s I (*I*) was calculated to measure the parent spatial distribution of each variable, based on a queen contiguity-based spatial weight matrix and the null hypothesis of complete spatial randomness [45]. Values range between −1 and 1. A stronger, positive Moran’s I indicated spatial clustering, those closer to zero indicated spatial randomness, and negative values indicated dispersion of like rates. *p* values conveyed significance, and the magnitude of the clustering index was used to compare variables. Anselin local Moran’s I was used to assess the underlying patterns of spatial homogeneity at the observation level to find high–high clusters, low–low clusters, and spatial outliers (high values in a low cluster, low values in a high cluster) [46]. As some degree of clustering will naturally exist even in the case of complete spatial randomness, a Monte Carlo test with 999 random permutations was used to increase precision by adjusting the range of possible pseudo-*p* values. Variables were compared visually by mapping significance in ArcGIS Pro, version 18 (Esri, inc.; Redlands, CA, USA) [47].

### 2.5. Correlating Human-Biting Tick Submissions and Human Disease

Bivariate correlations were used to quantify the relationship between the rates of human-biting *I. scapularis* submissions (adult, nymph, infected) and the rates of human disease. Correlations were based on Spearman’s Rho (ρ) and were made for point observations (each town per year) and aggregated sums (the total rate for each town 2015–2021). An analytic estimation of standard error was used to obtain 95% confidence intervals (Appendix A) [40]. The analysis focused on a presence-only dataset, requiring both tick submissions and presence of human disease to be included. Therefore, point observations (a town in a given year) that did not submit human-biting *I. scapularis* or report human disease were omitted. Correlations were then plotted. To account for the wide range of rates and skewed distribution, a log transformation was applied for visualization. Scatterplot axes therefore depict log10 (1+ rate) for each variable. Years were separated by colors in point observation scatterplots to display inter-annual variation.

## 3. Results

### 3.1. Descriptive Information of Tick Submissions

A total of 24,444 ticks were submitted to TickReport from 2015 through 2021 (Figure 2a). Most (95%) were collected from humans. A few (5%) were removed from canines, felines, equines, goats, and household objects. The number of submissions varied by year and included 15 species of human-biting ticks, the majority of which were *I. scapularis* (19,405), *Dermacentor variabilis* (3057), and *Amblyomma americanum* (461). Of the 19,405 human-biting *I. scapularis* submitted, 77% were adults, 22% were nymphs, and 1% were larvae (Figure 2b). 

Ticks spent an average of 20.3 h of victim-reported time attached to their human hosts (max = 4.1 days, min = 1 min.), with wide variation among individuals (Standard deviation (SD) = 20.1 h). The distribution was shown to significantly differ by tick life stage. Adults (mean = 20.98 h) spent an average of 3 more reported hours attached (*x*^2^ = 99.19, *p* < 0.001) than nymphs and larvae (mean_nymphs_ = 17.28 h, mean_larvae_ = 17.33 h). The proportion of engorged/replete ticks by life stage, as determined by the laboratory, was around 10% for each life stage (10.0% adults, 10.8% nymphs, and 9.7% larvae: *x*^2^ = 2.17, *p* = 0.338). Victims 60 years and older reported much longer attachment (mean_adults_ = 35 h, mean_nymphs_ = 27 h). The laboratory-determined engorgement rates of adult ticks were also twice as high for adults 60 years and older (17%) and for children 10 years and younger (20%). Considering age further, older victims submitted more adults (mean = 41 years) than nymphs (mean = 34 years) and larvae (mean = 26 years), H = 13.13, *p* = 0.001. Adults were primarily found attached to the head, neck, and back. More nymphs and larvae were found on the lower extremities (Appendix A).

### 3.2. Pathogen Presence in Tick Submissions

The cumulative prevalence of *A. phagocytophilum* was 8.00% [7.56, 8.45] in adults and 4.59% [3.97, 5.26] in nymphs (Figure 3a,c). *B. microti* prevalence was 8.37% [7.93, 8.84] in adults and 6.38% [5.66, 7.16] in nymphs (Figure 3b,d). A total of 1.27% of adults and 0.41% of nymphs were coinfected with *A. phagocytophilum* and *B. microti*. Pathogen prevalence was relatively stable across the study, though a small, non-significant increase was seen with *B. microti* in adult ticks from 2015 to 2016. The consistency in prevalence estimates was illustrated by negligible year-to-year variation and overlapping confidence intervals about prevalence estimates (Figure 3). Some geographical variation was seen, but estimates were similar in areas with large numbers of submissions. Interpretation of prevalence was difficult in locations with few submissions, particularly in the Suffolk (total of 19 nymphs, 66 adults) and Nantucket counties (55 nymphs, 18 adults) (Table 1).

### 3.3. Spatial Dynamics of Tick Submissions

Most victims submitting ticks (96%) reported a Massachusetts town of residence. The town where the bite was reported matched the victim’s town of residence in 82% of instances. The county of probable bite matched the county of residence in 92% of submissions. In total, 343 of 351 (98% of) towns submitted ticks during the study (Table 2). Submission rates were consistent from year to year from individual towns (towns that had a high rate of submissions generally did so across the 7 years, and towns that did not submit many generally did so across years). Total coverage by year tracked with the number of total submissions (Figure 2a, Table 2). Significant spatial clustering was observed with total (*I* = 0.13, *p* < 0.001) and infected submissions (*I*_bab_ = 0.06, *p* = 0.017; *I*_ana_ = 0.094, *p* < 0.001). Overlap was shown with clustering of rates for tick submissions, with high–high clusters found in the western and southeast parts of the state. Low–low clusters were found along Suffolk County, through the northeast region. An expected high–high cluster was also found in the towns around the testing laboratory (Figure 4a,b). 

### 3.4. Human Disease Reporting

A total of 8272 anaplasmosis and 5115 babesiosis cases were reported in Massachusetts over the study period. The yearly incidence of each disease doubled from 2015 to 2021 when considering all cases (Figure 5). Confirmed and probable counts were relatively stable for both diseases. Suspected counts, in comparison, were associated with consistent increases across time (ρ_bab_ = 0.96, *p* = 0.003; ρ_ana_ = 0.89, *p* = 0.01). When mapped spatially, the ratio of the rates of suspected cases to confirmed and probable cases (bab = 0.31, ana = 0.45) exhibited spatial randomness (*I*_bab_ = 0.025, *I*_ana_ = 0.022). Moderate variation was, however, seen between towns (mean_bab_ = 0.32, SD = 0.86; mean_ana_ = 0.96, SD = 2.1). This indicated that loss to follow-up and reporting practices varied by town, but clusters of systematically differing practices of reporting and clinical follow-up were not present.

Anaplasmosis was reported in 344 towns (98% coverage) and babesiosis in 300 (85% coverage). The yearly number of towns reporting each disease increased consistently, from 221 to 299 for anaplasmosis and 164 to 233 for babesiosis (Table 2). Town-level incidence rates were clustered to a similar degree with both high–high and low–low clusters present (*I*_bab_ = 0.59; *I*_ana_ = 0.66). Although these regions overlapped, there were locations where the magnitude and presence of disease differed greatly. Differences (up to 8–10× in some areas) were best illustrated by looking at discrepancies between disease rates at the county level (Table 3). The differences in the location and magnitude of significant clusters weakened with time as the spatial coverage of each disease increased (Table 2, Figure 4).

### 3.5. Seasonality of Human Disease and Tick Submissions

Seasonal human disease patterns differed for anaplasmosis and babesiosis (Figure 6). Babesiosis was almost entirely reported during the summer, peaking in July. A very small second peak was seen in November of each year. Negligible variation in the seasonality of these diseases was observed year to year. Babesiosis cases aligned almost perfectly one month following nymphal *I. scapularis* submission peaks (Figure 6b,d). Most (85%) nymphs were submitted from May through July with a consistent peak in June of each year and a second smaller peak present in October. Anaplasmosis, in comparison, regularly peaked in June, with a second prominent peak in November. This was more consistent with adult activity, as cases peaked in the summer, 1-month after spring adult activity, but aligned temporally with the November peak in adult submissions. Adult submissions followed a bimodal distribution, with more variation in the intensity and the onset of summer and fall peaks (Figure 6a,c). In general, adult submissions peaked in May (March through June) and in the fall, each November.

### 3.6. Quantifying the Correlation between Tick Submissions and Human Disease

The 2015–2021 aggregated rates of total, adult, nymph, and infected *I. scapularis* submissions were significantly correlated with human anaplasmosis and babesiosis case rates, each at a *p* < 0.001 (Figure 7, Table 4). The inclusion of infection status was not more informative than submission numbers alone. Correlations were also similar when assessed across years as point observations, and maintained the pattern shown with aggregated sums (Figure 8, Table 4). Aggregated correlations were stronger for each measure with anaplasmosis (ρ range: 0.708 to 0.830) than babesiosis (ρ range: 0.552 to 0.684), apart from those of nymphal submission rates, which were comparable (ρ_ana_ = 0.708, ρ_bab_ = 0.684). The relationships between babesiosis and adult and infected submission rates were less clear, and scatterplots for these correlations showed patterns with less apparent monotonicity. Moderate correlations (ρ_adult_ = 0.552, ρ_infected_ = 0.584) were, however, still present. As adults represented a larger proportion of total *I. scapularis* submissions, the pattern observed with total submission was like that of adult and *B. microti*-infected submissions (ρ = 0.597). Correlations with anaplasmosis, in comparison, were all very clear, and the patterns of adults, nymphs, and infected submissions did not greatly differ visually or quantitatively.

Mismatches (submissions occurring and disease absent, disease occurring and submissions absent) and outliers of high/low mismatched rates were present, though uncommon. Town population ranged from 69 to 674,272 (mean = 20,051, SD = 42,341). This was problematic, particularly for towns with fewer than 1000 residents. Solely considering these (*n* = 28), human disease and total tick submissions were not significantly correlated (ρ_ana_ = −0.21, *p* = 0.280; ρ_bab_ = −0.15, *p* = 0.440). The towns where anaplasmosis was not reported each had fewer than 1000 residents. Babesiosis was similarly not reported in poorly populated towns. However, many towns that did not report babesiosis were more populated.

## 4. Discussion

The CDC acknowledges that tick testing can provide useful information for research purposes; however, they do not recommend tick-bite victims or physicians pursue this service. This is due to the concern that many people who become sick are not aware of being bitten by a tick, and that testing may yield misleading results as (1) false positives arise from true negative results or strains nonpathogenic to humans; (2) infection in the tick is not necessarily indicative of disease transmission; and (3) a negative test can lead to complacency regarding symptom monitoring [7,36].

The results of the present study, however, show clear correlations between submission rates and total reported disease, which provide insights into human–tick encounters across space and time, and quantified problems with human disease reporting due to misclassification and loss to follow-up. Key differences between the two diseases were also identified. Babesiosis aligned well with nymphal submission numbers, but less so with adults. The seasonality of babesiosis also aligned almost perfectly with nymphal submissions, accounting for an allotted 1–4 weeks for incubation, seeking care, and diagnostics. Anaplasmosis was more correlated with adult submissions. The finding that anaplasmosis cases lagged behind submissions by one month in the summer but aligned perfectly with submissions in the fall is peculiar. This is unexplained, and warrants further investigation, but likely results from an aspect of human behavior not considered here.

This work builds on previous studies which have assessed the ability of passive surveillance to predict Lyme disease risk in emergent and endemic regions. A recent study in the neighboring state of Connecticut, an area endemic for Lyme disease, made the connection between passive human-biting tick surveillance and the spatio-temporal patterns of Lyme disease incidence [28]. Citizen science has also been used on a wider scale in the United States to gather information on the distribution and pathogen prevalence of human-biting ticks [25,26,27]. These methods have been applied in early warning systems to predict regions of emergent vector and *B. burgdorferi* presence [35]. The validation of these methods in emergent regions is particularly important, considering the expanding range of medically important ticks including *I. scapularis*, *A. americanum*, *D. variabilis*, and *Haemaphysalis longicornis* [3,48,49]. This work, however, represents the first extension of these relationships to endemic diseases other than Lyme disease. This is important, considering the increase in the spread of these diseases.

Spatial scale and aggregation schemes have varied with previous investigations. Some studies have focused on ZIP code tabulation areas, at town, county, or state level. Stakeholders will have more of an interest in one level or another, depending on objectives. The present study focused on town-level measures, which provided data on a fine spatial scale, allowing sub-county variability to be identified while maintaining case and tick submission confidentiality. Population was shown to be problematic regarding correlations, especially in towns with fewer than 1000 residents. These are areas that may better be targeted with an active entomological survey. The modifiable unit area problem must also be considered when interpreting results. Bayesian approaches have been introduced to mitigate issues associated with these spatial measures [50,51]. Measurement consistency is one of the most important pillars of an effective surveillance program, and work should be carried out to better understand how to best describe tick-borne disease risk across space [52,53].

Little year-to-year change in pathogen prevalence was noted throughout the 7 years studied, though they represent an increase on past measurements. Prevalence estimates in Massachusetts from 2006 to 2012 were 4.6% for *A. phagocytophilum* and 1.8% for *B. microti* [30]. From 2015 to 2021, however, *A. phagocytophilum* and *B. microti* infection rates were higher, and remained around 8–9% in adults and 4–6% in nymphs. Although interpretation was difficult in areas with few submissions, well-represented areas gave reliable estimates that showed similarity across space. These prevalence estimates were remarkably like the findings of recent active surveillance in this region. For example, an overlapping (2013–2019) survey performed across the United States reported nymphal (adult) *I. scapularis* prevalence of 5.76% (8.07%) for *A. phagocytophilum* and 5.69% (3.53%) for *B. microti* in New England [54]. Similar results were shown for flagged ticks in the neighboring state of Connecticut: 4.0% (7.8%) for *A. phagocytophilum* and 6.8% (8.6%) for *B. microti* [55]. Testing of human-biting ticks has in general been previously noted to lack the overall power to assess pathogen prevalence in ticks, which is found with active flagging/dragging efforts [56]. The accordance between these results suggests that passive and active surveillance may be used together to assess entomological measures of risk, with active surveillance particularly useful in filling in gaps where passive submissions are lacking. Future work is needed to assess the accordance of these approaches. Accounting for pathogen prevalence in submitted ticks did not improve the relationships seen, suggesting that submission rates alone can document the human–tick interactions that are predictive of disease. This is most likely due to Massachusetts being an endemic region. Incorporating tick pathogen prevalence is going to be more important in emergent regions, where prevalence will vary to a greater extent across space.

Hard ticks (such as *I. scapularis*) feed for several days, and it is well understood that the risk of pathogen transmission increases with attachment time [57,58]. The minimum attachment time required for transmission and typical transmission dynamics of most tick-borne pathogens are, however, poorly studied. Studies in animal models have demonstrated that *A. phagocytophilum* injection into the host is possible within 24 h of attachment, although sustained transmission likely requires 24–36 h of feeding [59,60,61]. *B. microti* transmission, in comparison. has been shown to require closer to 36–54 h, due to a mechanism of feeding-induced sporogony [62,63,64]. Surveillance from the Massachusetts Department of Health has previously demonstrated that clinically diseased humans with anaplasmosis are twice as likely to have reported knowledge of a recent tick bite than those with babesiosis [8,10]. In general, adults are larger and more likely to be noticed by bite victims, leading to higher rates of removal and shorter feeding duration. These data, in addition to the alignment of human cases with tick submissions in the present study, are evidence to support a longer feeding time required to transmit *B. microti* than *A. phagocytophilum*.

The findings that engorgement rates and attachment times were longer for victims 60 years and older and 10 and under is also like that which has been previously reported in human-biting tick studies and coincides with reporting by the Massachusetts Department of Health, which finds more than half of clinical cases in humans 60 years and older. This re-asserts the need to target community health education measures at the older population. Increased engorgement rates in younger children also warrant outreach to parents and guardians on mitigating this risk. Victim-reported attachment times for nymphal bites were shorter than adults, despite laboratory-determined engorgement rates being similar. Self-reported attachment time is known to underestimate true nymphal feeding time and is unreliable for individual risk assessment [65,66]. Quantitative means have been used to more accurately approximate attachment duration, based on the ratio of the width of the scutum to the coxal gap (coxal index) or the ratio of the width of the scutum to the length of the idiosoma (scutal index) [67,68]. Better understanding of transmission dynamics is an important future direction to aid in risk assessment and diagnostics, as well as direct recommendations for the development and use of personal protection [58,69]. TickReport does not follow up on bite victims that submit ticks. This line of study represents an opportunity for future studies to further investigate this by linking tick submission programs with clinical follow-up and integrating quantitative measures to analyze attachment time.

A limitation of the anaplasmosis findings is that the molecular assay used to identify *A. phagocytophilum* does not differentiate between the human active genetic variant (Ap-ha), which causes anaplasmosis and is maintained in the white-footed mouse (*Peromyscus leucopus*) and eastern chipmunk (*Tamias striatus*), and the deer variant (Ap-v1), which is maintained in the white-tailed deer (*Odocoileus virginianus*) and is not pathogenic to humans (Appendix A) [4,9,70]. Assays differentiating these variants have been developed and used to compare the relative abundance of each variant to human disease presence [71,72]. These assays are not, however, commonly employed in tick surveillance systems. It may be reasonable to assume that the relationships identified here hold, and that disease increases are possibly explained by a spatial shift in the relative density of Ap-ha to Ap-v1 (in addition to increased awareness and reporting). Future work is, however, needed to look further into this relationship, and the implementation of these assays should be further considered.

Limitations inherent to passive tick surveillance and passive surveillance also apply [31,33,34]. The data used focused on a presence-only dataset. The coverage of tick submissions also varied across space and time. This could have been due to variation in program knowledge, public and physician perception of tick testing, and socioeconomic factors that would influence submissions. Further, low population numbers impacted the relationship observed between disease and submissions. Submissions were also strongly skewed towards adult ticks, as they are easier to identify. Finally, submission information such as attachment time and where/when people were at the time of encounter are often limited to a window of opportunity, and may not be precise.

Each form of surveillance is subject to limitations. Human disease reporting is essential for understanding who is getting sick. The present study, however, showed the magnitude to which surveillance is affected by spatial-varying loss to follow-up. Further, travel information is rarely available, socioeconomic inequity affects the likelihood of seeking healthcare and health outcomes, and disease is tied to the place of residence, thus restricting the assessment of risk in areas of recreation and vacation. Case studies are poor risk indicators of babesiosis and anaplasmosis, due to their rare incidence and lack of distinctive clinical signs, such as erythema migrans, which manifest with Lyme disease [73]. Active surveillance is important in describing the density and pathogen prevalence in host-seeking ticks (the main predictors of acarological risk), especially with nymphs, as they are less accounted for with passive testing. Questing behavior is affected by many environmental conditions, which impacts collection. In addition, tick abundance does not necessarily translate into increased levels of human disease, due to local differences in human behavior and recreation which may affect exposure risk, the possibility of microhabitat occurrence, and the use of personal protection measures that may mitigate risk in an otherwise high-risk area [74]. Wide-scale application is also difficult, due to high resource and time demands. Seroprevalence studies with sentinel species can provide sensitive and long-term data, but are affected by scale and selection bias. Added to this, we still do not fully understand many of the ecological relationships which affect ticks and tick-borne pathogens.

## 5. Conclusions

Integrative approaches to tick-borne disease surveillance are needed to address the complexity of tick ecology and disease etiology. The results of the present study show how measures obtained from passive surveillance of human-biting ticks and their pathogen testing correlate with human babesiosis and anaplasmosis incidence. Human case data and active flagging surveillance are important for learning about who is getting sick and the activity of ticks and their pathogens. Tick testing and passive surveillance measures have the potential to fill in gaps between these two methods, connecting human behavior and acarological risk, to inform public health intervention. Future studies are needed to understand the agreement between risk measures of active and passive tick surveillance and identify ways in which data collection from passive tick testing can be improved to better inform public health activity.

## Figures and Tables

**Figure 1 microorganisms-11-01418-f001:**
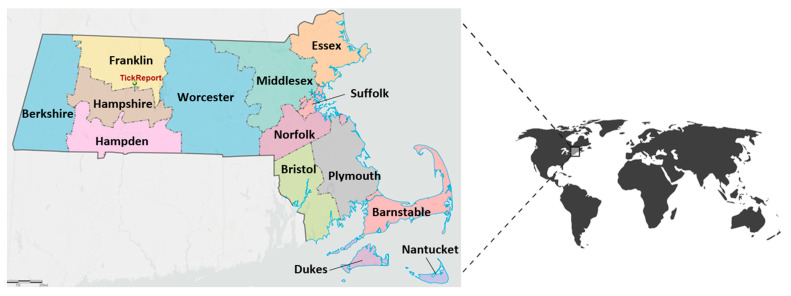
The study location (Massachusetts, USA) is shown relative to the world map. Counties are shown in black print. TickReport in Amherst, MA, where tick testing occurred, is noted in red print with a green pin.

**Figure 2 microorganisms-11-01418-f002:**
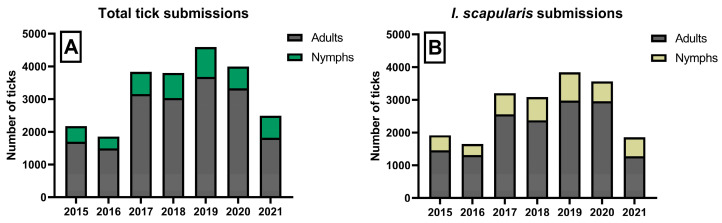
Yearly trends in (**A**) total adult and nymphal tick submissions; (**B**) adult and nymphal I. scapularis submissions to TickReport, 2015–2021.

**Figure 3 microorganisms-11-01418-f003:**
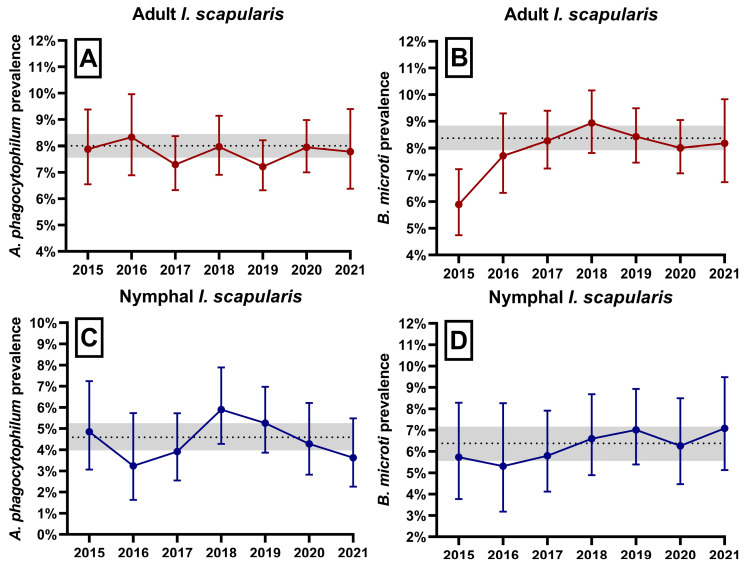
Annual prevalence of *A. phagocytophilum* and *B. microti* in (**A**,**B**) adult; and (**C**,**D**) nymphal *I. scapularis.* Error bars represent 95% confidence intervals around yearly prevalence measures. Gray horizontal, dotted lines and surrounding filled regions indicate cumulative prevalence 2015–2021 and associated 95% confidence intervals.

**Figure 4 microorganisms-11-01418-f004:**
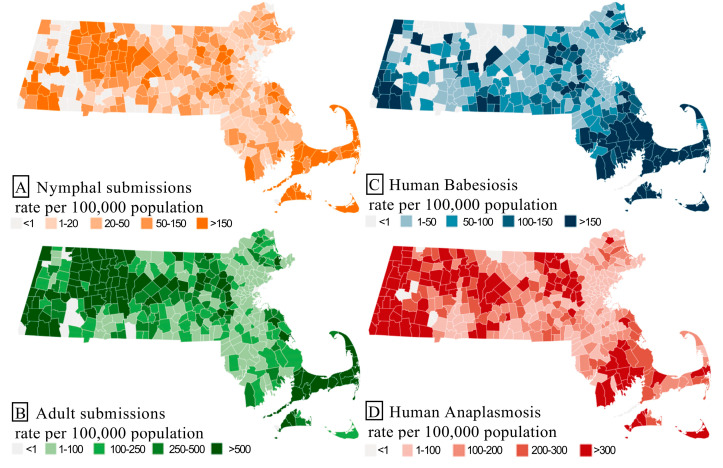
2015–2021 town-level rates per 100,000 population of (**A**) nymphal *I. scapularis* submissions; (**B**) adult *I. scapularis* submissions; (**C**) human babesiosis; (**D**) human anaplasmosis.

**Figure 5 microorganisms-11-01418-f005:**
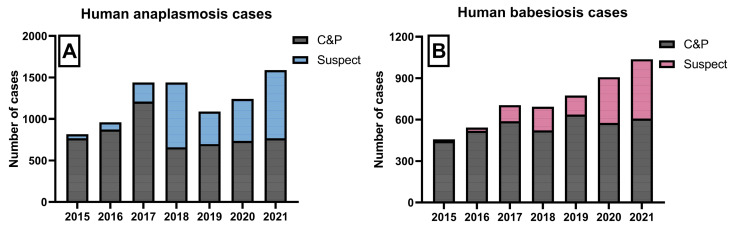
Yearly reported confirmed and probable (C and P) and suspected cases of (**A**) human anaplasmosis; and (**B**) human babesiosis.

**Figure 6 microorganisms-11-01418-f006:**
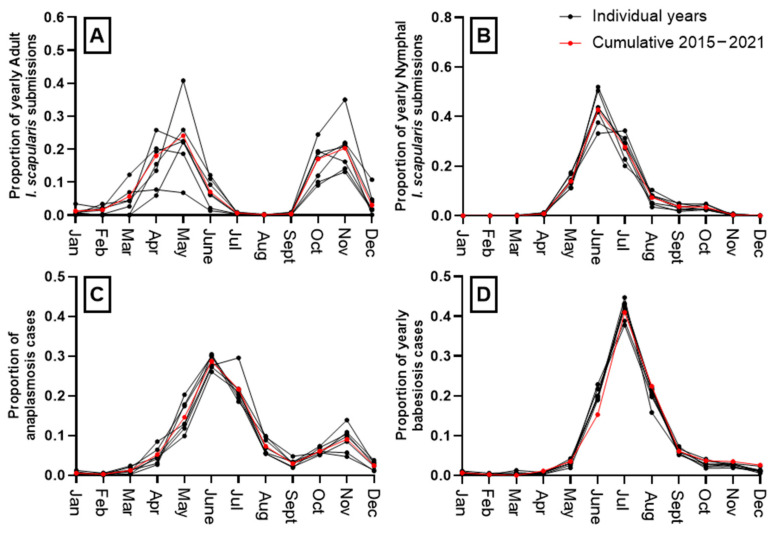
Seasonal trends of (**A**) adult *I. scapularis* submissions; (**B**) nymphal *I. scapularis* submissions; (**C**) human anaplasmosis cases; (**D**) human babesiosis cases. Years 2015–2021 are individually represented in black and cumulatively shown in red. Months are abbreviated where noted: January (Jan), February (Feb), March (Mar), April (Apr), May, June, July (Jul), August (Aug), September (Sept), October (Oct), November (Nov), December (Dec).

**Figure 7 microorganisms-11-01418-f007:**
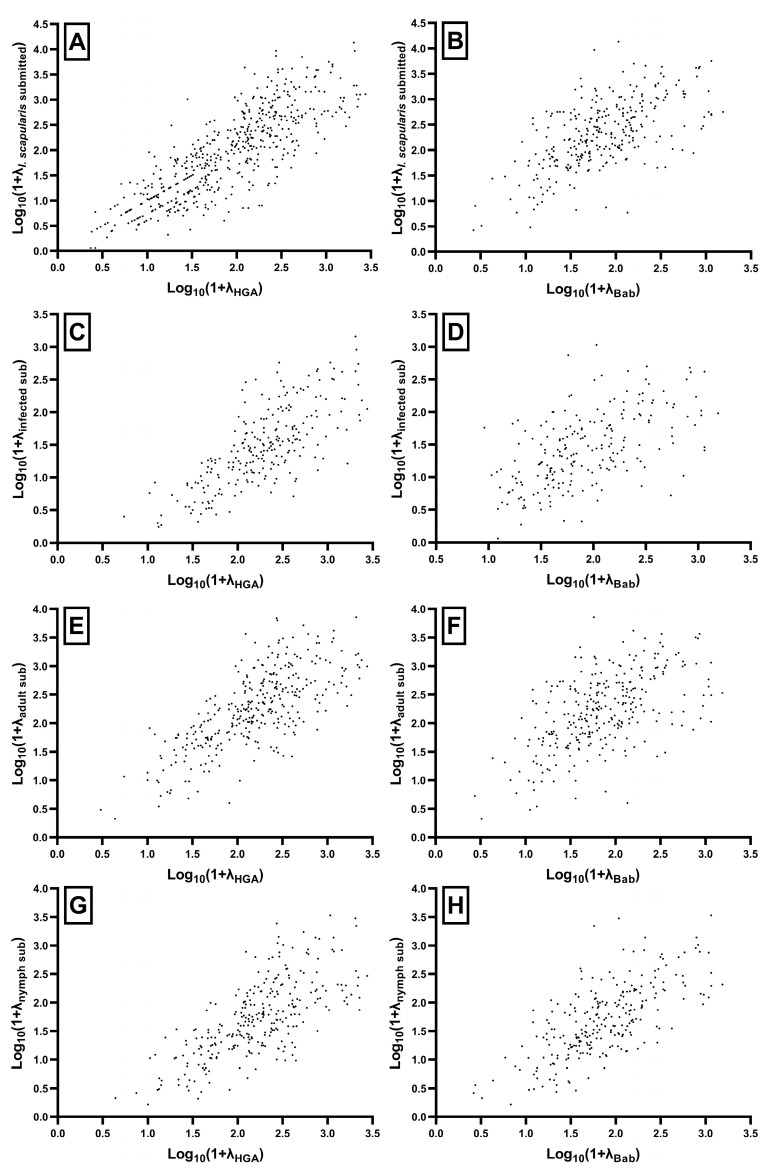
2015–2021 aggregated correlations of *I. scapularis* submissions and human disease: (**A**,**B**) Total submissions; (**C**,**D**) Infected submissions; (**E**,**F**) Adult submissions; (**G**,**H**) Nymphal submissions. Points represent individual towns, with a log10-transformation applied. λ = rate per 100,000 town population; HGA = Human granulocytic anaplasmosis (anaplasmosis); Bab = Babesiosis.

**Figure 8 microorganisms-11-01418-f008:**
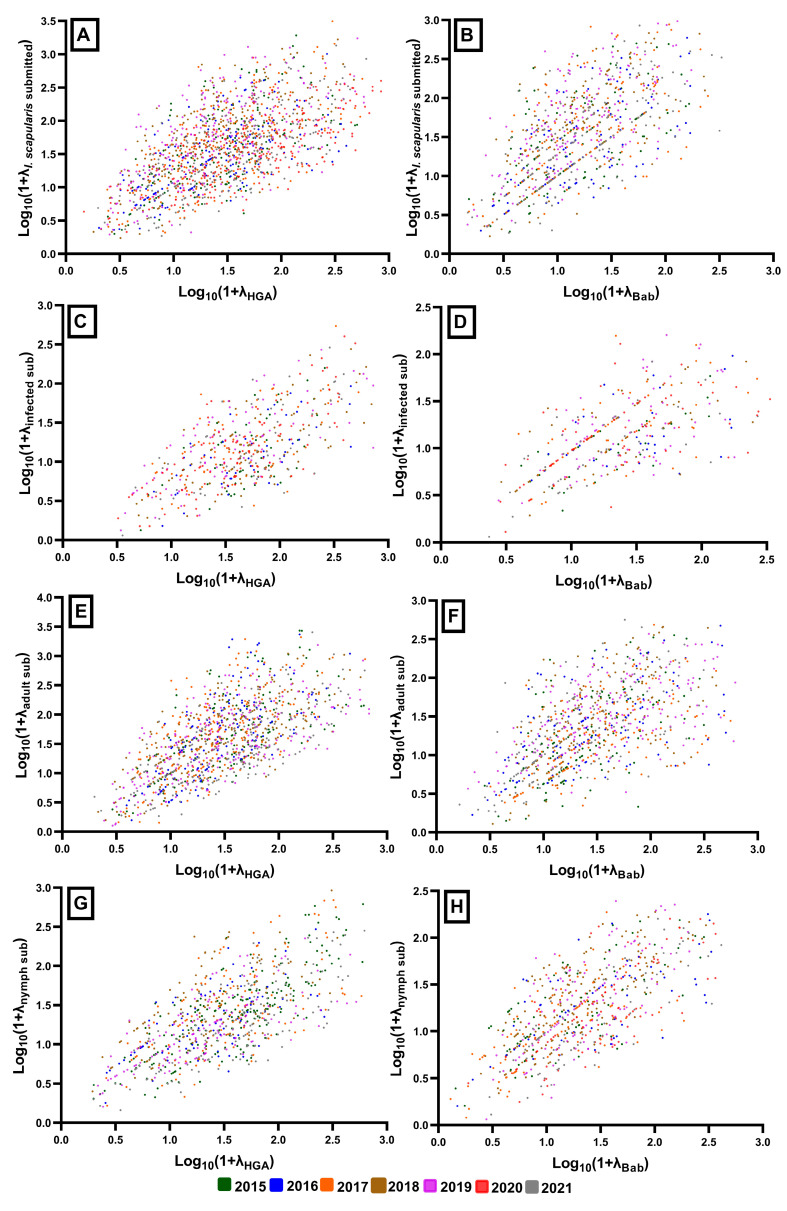
Point correlations of *I. scapularis* submissions and human disease: (**A**,**B**) Total submissions; (**C**,**D**) Infected submissions; (**E**,**F**) Adult submissions; (**G**,**H**) Nymphal submissions. Points represent individuals in each year, with a log10- transformation applied. Colors are used to designate years, and λ = rate per 100,000 town population. HGA = Human granulocytic anaplasmosis (anaplasmosis); Bab = Babesiosis.

**Table 1 microorganisms-11-01418-t001:** County-level nymphal (NIP) and adult (AIP) prevalence, cumulative 2015–2021.

County	Nymphs Tested	NIP		Adults Tested	AIP	
		*A. phag* ^1^	*B. microti*		*A. phag*	*B. microti*
Barnstable	1041	3.46%	9.41%	3793	6.20%	9.52%
Berkshire	104	7.69%	0.01%	594	13.80%	8.08%
Bristol	192	5.21%	8.85%	359	5.57%	5.29%
Dukes	270	6.30%	7.78%	147	8.16%	6.80%
Essex	142	4.23%	7.75%	685	9.78%	11.53%
Franklin	378	6.35%	2.12%	1512	8.27%	6.22%
Hamden	76	6.58%	3.95%	388	3.87%	6.70%
Hampshire	386	3.37%	3.37%	1636	7.03%	6.00%
Middlesex	544	4.41%	5.70%	2319	8.80%	8.58%
Nantucket	55	7.27%	5.45%	18	11.11%	5.56%
Norfolk	190	4.21%	6.84%	671	6.70%	7.00%
Plymouth	322	4.66%	9.00%	900	7.11%	9.11%
Suffolk	19	5.26%	0.00%	66	3.00%	1.52%
Worcester	256	4.30%	2.35%	1488	8.60%	7.53%

^1^*A. phagocytophilum*.

**Table 2 microorganisms-11-01418-t002:** Spatial coverage of human disease and tick submissions by number of towns (percent coverage of 351 total Massachusetts towns).

Year	Anaplasmosis	Babesiosis	Tick Submissions
2015	221 (63%)	164 (47%)	244 (63%)
2016	242 (69%)	156 (44%)	239 (69%)
2017	289 (82%)	198 (56%)	284 (82%)
2018	272 (77%)	191 (54%)	305 (77%)
2019	271 (77%)	206 (59%)	306 (77%)
2020	281 (80%)	200 (57%)	305 (80%)
2021	299 (85%)	233 (66%)	268 (85%)
Total Representation	344 (98%)	300 (85%)	343 (98%)
7-Year Change	+35%	+42%	+10%

**Table 3 microorganisms-11-01418-t003:** County-level incidence of human disease and submission rates per 100,000 population, cumulative 2015–2021.

County	Anaplasmosis Rate	Babesiosis Rate	Adult Sub Rate	Nymphal Sub Rate
Barnstable	252.87	333.23	1601.96	419.71
Berkshire	820.91	104.07	379.78	65.24
Bristol	160.55	177.36	55.14	28.75
Dukes	461.59	685.09	660.80	1127.25
Essex	52.64	31.88	76.98	15.82
Franklin	421.19	40.85	1960.89	480.36
Hamden	47.96	26.67	75.92	15.06
Hampshire	223.98	41.34	945.30	233.99
Middlesex	93.77	36.27	130.90	30.87
Nantucket	540.57	1151.36	105.30	329.96
Norfolk	66.80	48.20	81.67	21.62
Plymouth	225.24	196.24	154.80	52.73
Suffolk	11.44	10.18	7.04	1.89
Worcester	112.01	37.37	157.29	26.58

**Table 4 microorganisms-11-01418-t004:** Point and aggregated correlations of tick submissions and human disease rates using Spearman’s Rho ^1,2^.

Measure	2015	2016	2017	2018	2019	2020	2021	Aggregated
Anaplasmosis								
All sub	0.655	0.668	0.662	0.678	0.625	0.695	0.720	0.830
Inf sub	0.541	0.672	0.590	0.683	0.722	0.707	0.650	0.732
Adult	0.700	0.727	0.625	0.688	0.670	0.701	0.728	0.713
Nymph	0.791	0.622	0.617	0.660	0.735	0.684	0.759	0.708
Babesiosis								
All sub	0.584	0.667	0.594	0.634	0.633	0.614	0.712	0.597
Inf sub	0.773	0.407	0.642	0.722	0.517	0.609	0.710	0.584
Adult	0.591	0.596	0.586	0.581	0.567	0.580	0.684	0.552
Nymph	0.661	0.644	0.667	0.753	0.675	0.722	0.753	0.684

^1^ All correlations are significant at *p* < 0.001. ^2^ Refer to Appendix A for 95% confidence intervals for rho for each relationship.

## Data Availability

TickReport data are made available in real time and may be accessed at TickReport.com/stats. Human case data are not made available, due to privacy restrictions. Data were obtained from the Massachusetts Department of Health (MDPH), and may be made available upon reasonable request with the permission of the MDPH. Requests should be submitted by completing the Surveillance Data Request Form accessed here: https://www.mass.gov/doc/surveillance-data-request-form/download, accessed 18 February 2023.

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
