# Peer review of "Human-Biting *Ixodes scapularis* Submissions to a Crowd-Funded Tick Testing Program Correlate with the Incidence of Rare Tick-Borne Disease: A Seven-Year Retrospective Study of Anaplasmosis and Babesiosis in Massachusetts"

_microorganisms, 2023, doi:10.3390/microorganisms11061418_

Round 1
Reviewer 1 Report
Dear Editor,
The paper "Human-biting Ixodes scapularis submissions to a crowd-funded tick testing program correlate with the incidence of rare tick-borne disease: A seven-year retrospective study of anaplasmosis and babesiosis in Massachusetts" is excellently written and carried out.
It might be quite interesting to the scientific community after some modest adjustments. Although a study of this kind has inherent limitations, these constraints were effectively discussed during the discussion.
Numerous scientific names are not in italics, please check it in all the manuscript.
Regarding to conclusions, this section might, in my opinion, be condensed to solely include the authors' thoughts, excluding quotes from other research.
Author Response
The paper "Human-biting Ixodes scapularis submissions to a crowd-funded tick testing program correlate with the incidence of rare tick-borne disease: A seven-year retrospective study of anaplasmosis and babesiosis in Massachusetts" is excellently written and carried out.
It might be quite interesting to the scientific community after some modest adjustments. Although a study of this kind has inherent limitations, these constraints were effectively discussed during the discussion.
- Numerous scientific names are not in italics, please check it in all the manuscript.
The authors appreciate the reviewer raising this issue and apologise with the issues to formatting arising at the time of submission. This issue has been resolved in all occurrences in the manuscript.
- Regarding to conclusions, this section might, in my opinion, be condensed to solely include the authors' thoughts, excluding quotes from other research.
The authors thank the reviewer for this great point and have amended the conclusions to address this. New information is not presented and therefore citations have been removed.
Reviewer 2 Report
All scientific names are in italics. In all the text, they are not. An introduction would be a concise, idem discussion.
Line 39-42 Mention which stage of the species is the one that transmits the disease since the highest proportion of ticks found was in adults, where nymphs and adults were found. For example, this is described in bovines. The larvae are those that transmit B. bovis, and the nymphs B. bigemina. In the case of anaplasmosis, it is adult ticks or mainly blood-sucking flies – A. marginale.
Line 190-193 Not clear
It is a good case study. However, it would have been interesting to analyze the adaptation and distribution conditions of the I. scapularis tick. That is, climate change is correlated with the presence of tick populations and the presence of diseases over the years.
It would be essential to know what social status the ticks were removed from and in what areas... to be able to correlate why or where the ticks come from?
I think another approach can be given to the data collected and the information published would be more valuable.
No comments.
Author Response
- All scientific names are in italics. In all the text, they are not. An introduction would be a concise, idem discussion.
The authors apologise for formatting errors when submitting the manuscript and have ensured italics are used where needed in the revisions.
- Line 39-42 Mention which stage of the species is the one that transmits the disease since the highest proportion of ticks found was in adults, where nymphs and adults were found. For example, this is described in bovines. The larvae are those that transmit B. bovis, and the nymphs B. bigemina. In the case of anaplasmosis, it is adult ticks or mainly blood-sucking flies – A. marginale.
The authors thank the reviewer for the comment. Human babesiosis and anaplasmosis in this region are caused by what is outlined in lines 40-45. References 6 and 7 provide extensive background on this. Nymphs and adults both transmit disease, though infection prevalence increases from the nymphal stage as they feed on another host. The skewness towards adult submissions is based on what bite victims see and is not a proxy for density measures. This is touched on in the discussion, lines 437, 454.
- Line 190-193 Not clear
The authors thank the reviewer for the comment regarding the clarity of section 3.2. However, as this is a description of the pathogen prevalence results from the study, also visualised in figure 2, there is nothing that is revisable.
- It is a good case study. However, it would have been interesting to analyze the adaptation and distribution conditions of the I. scapularis tick. That is, climate change is correlated with the presence of tick populations and the presence of diseases over the years.
The authors than the reviewer for the comment. Please refer to https://doi.org/10.1002/ecm.1572, where these experiments and modeling were done. This paper keeps its focus on where passive surveillance fits within the realm of tick borne disease surveillance, with the purpose of documenting human-tick encounters. By documenting bites through this system, we are able to see trends in establishment, refer to similar studies in emerging regions 10.1093/jme/tjy030, of both the vector and pathogen. This has been done in endemic regions and this work assessed whether this could extend beyond Lyme disease with less common tick-borne diseases. Adaptation and distribution are outside of the scope of this study.
- It would be essential to know what social status the ticks were removed from and in what areas... to be able to correlate why or where the ticks come from? I think another approach can be given to the data collected and the information published would be more valuable.
The authors thank the reviewer for the comment. Section 3.1, paragraph 2, goes into detail on the demographics of victims, considering age and sex. Beyond that, however, remains outside of the scope of the present study, requires enough where it may be its own focus, and has been recently done. Refer to https://doi.org/10.3390/ijerph20054306, where predictors of tick submissions are analysed, eg socioeconomic factors. The purpose of this manuscript was to assess the relationship between passive tick testing submissions and human disease for future work which may assess the accordance of acarological risk measures between active and passive means, under the assumption that if passive submissions are not associated with human disease, this work would not be meaningful.
Reviewer 3 Report
Please check the attached file

Author Response
The authors analyzed several years of human biting I. scapularis submissions to TickReport alongside human babesiosis and anaplasmosis incidence in a department of health.
- Although it is noted that they have done extensive work correlating information from the database, the authors do not seem to be very clear about the usefulness of this information (Lines 26 / 453) which makes it unclear to me as a reviewer as well.
The authors thank the reviewer for raising this issue with clarity. The conclusion section has been revised to better address this issue and make this more clear.
- Line 36: Start the sentence by introducing babesiosis and then anaplasmosis, as this is how you detailed them in later sentences
The authors agree with the reviewer’s point and have amended the sentence to begin with babesiosis.
- Line 42: There are several Anaplasma species that can cause anaplasmosis
The authors thank the reviewer for raising this question regarding the aetiology of human granulocytic anaplasmosis (HGA). HGA is caused by A. phagocytophilum. It has been noted that other Anaplasma spp. have been identified in humans, 10.3201/eid1606.090175, but the concern in New England is A. phagocytophilum, transmitted by I. scapularis.
- Line 60: Studies have also been conducted in domestic dogs and cats (and wildlife) to broaden the scope of vector-borne diseases research in humans. Please modify this sentence. Line 80: Lyme is not a mandatory reportable disease?
The authors appreciate the reviewer feedback. In the following sentence introducing passive surveillance, wildlife surveillance has been added to companion animal species for surveillance efforts.
- Line 103: Italics on I. scapularis. Please check throughout the manuscript for all scientific names.
The authors appreciate the reviewer raising this issue and apologise with the issues to formatting arising at the time of submission. This issue has been resolved in all occurrences in the manuscript.
- Figure 1: I recommend keeping Figure 1B and 1C in a separated figure for better understanding with the text.
The authors thank the reviewer for the input. Figure 1B and 1C are now their own figure (Figure 3). In addition to figure 1A, an additional panel showing total tick submissions (in addition to just I. scpaularis) has been included (Now Figure 2, A and B).
- Line 184: “The cumulative prevalence of A. phagocytophilum in was 8.00%” Remove the word“ in” before the prevalence.
The authors are appreciative of the reviewer’s attention to detail and apolgise for the typo in the original manuscript. This has been revised accordingly.
- Figure 2: What is the difference between “pathogen prevalence” (Line 187) and “infection prevalence” (Lune 190)?
The authors thank the reviewer for raising this point on consistency and clarity. Throughout the manuscript, this is now solely referred to as pathogen prevalence.
- Line 199: Nowhere in the M&M is there any information about the country in which this study was conducted. Even if it was conducted in the U.S.A., it should be clarified
The authors thank the reviewer for raising this point. For clarity, this information has been included in section 2.4 as well as in the introduction. A new figure (figure 1) has been included to show the study location relative to the world map.
- Figure 3: A small map of the globe would be useful to orient the readers as to which site these images refer to. Add the State to which these maps refer. Also, the letters A-D representing each map are confusing, as they can be mistaken with the genus of a species. Please add a parenthesis or some other clarifying element.
The authors thank the reviewer for pointing out how to make the study site clearer for those unfamiliar with the area. A new figure has been included, including a county-level map that indicates the location of the tick testing laboratory. For town-level maps, the indicating letters have been changed to be contained within boxes instead of the previous format to avoid confusion with a species genus.
- Line 223: “A total of 8,272 anaplasmosis and 5,115 babesiosis total cases” avoid repetition of words
The authors thank the reviewer for raising this point on clarity and have removed the second “total” to improve flow.
Reviewer 4 Report
Background
This is a good manuscript with some important findings. It extends work that has been done with Lyme and does a good job of calling attention to the too-neglected Anaplasma. I think the authors missed an opportunity to highlight the results of FIG2B, but overall this is a very solid manuscript on an increasingly important topic. Please do italicize genus and species names-I count at least 20 instances where this was not done.
“two slightly rarer diseases caused by I. scapularis”
A minor point here-the disease is not caused by I scapularis, but by anaplasma/babesia. It is transmitted by I scapularis.
“Anaplasmosis is caused by the rickettsial bacterium”
I never liked calling these rickettsial bacteria. Yes they are members of the Rickettsiales, but I think its important to highlight Anaplasma is part of anaplasmataceae, not rickettsiaeceae. I would prefer you remove the word rickettsial
“hospitalizing one fourth of symptomatic cases”
I would say one fourth of reported cases. Anaplasma surveillance is not very good in general, and it is likely there are asymptomatic cases that go unreported.
Methods
“and confirmed cases had laboratory-confirming diagnostics 120 (such as PCR).”
I would like it if you listed out the lab confirming diagnostics here-is immunohistochemistry included? IgG based conversions? Cell culture?
“Bivariate correlations were used to quantify the relationship between the rates of hu- 147 man-biting I. scapularis”
I notice that many times you do not italicize I scapularis (twice in this paragraph alone). Please search the document and properly italicize this.
Results
“e I. scapularis 164 (19,405), Dermacentor variabilis (3,057), and Amblyomma americanum (461). Of the 165 19,405 human-biting I. scapularis submitted, 77% were adults, 22% were nymphs, and 1% 166 were larvae (Figure 1a).”
As above, please italicize genus/species
“Adults (Mean = 169 20.98 hours) spent an average of 4 more reported hours attached than nymphs and larvae 170 (Meannymphs = 17.28 hours, Meanlarvae = 17.33 hours).”
You could easily prove this is significantly different-consider doing so. Depending on data a t test should be sufficient. Please also do so for other findings where you claim differences exist.
“Ticks spent an average of 20.3 hours of victim-reported time attached to their human 168 hosts. The distribution was shown to significantly differ by tick life stage. Adults (Mean = 169 20.98 hours) spent an average of 4 more reported hours attached than nymphs and larvae 170 (Meannymphs = 17.28 hours, Meanlarvae = 17.33 hours). The proportion of engorged/replete 171 ticks by life stage, as determined by the laboratory, was around 10% for each life stage. 172 Victims 60 years and older reported much longer attachment (Meanadults = 35 hours, 173 Meannymphs = 27 hours). The laboratory-determined engorgement rates of adult ticks were 174 also twice as high for adults 60 years and older (17%) and for children 10 years and 175 younger (20%). Further considering age, older victims submitted more adults (Mean = 41 176 years) than nymphs (Mean = 34 years) and larvae (Mean = 26 years). Adults were primar- 177 ily found attached to the head, neck and back while nymphs were mainly removed from 178 the lower extremities.”
You have all this great data. I would like to see it encapsulated in a table to help the readers.
“Figure 2B”
The prevalence increases here. This is fascinating-given that this was PCR derived, I have some faith that this is a true increase.
“Table 1. County-level nymphal (NIP) and adult (AIP) infection prevalence, cumulative 2015-2021”
I think the paper could benefit from a county level map of MA. I am interested in the study but not from the Northeast and I had to stop and go lookup counties and cities to follow this. Inserting a map could help (possibly earlier in the paper).
“The yearly number of towns reporting each disease increased stepwise,”
I don’t like the use of stepwise here. I think you might mean consistently?
“Seasonal disease patterns differed for both diseases, but little inter-annual variation 245 was seen for each.”
I do not think this is accurate. Consider your figure 2B. There is significant annual variation in prevalence of anaplasmosis. Figure 4 simply shows there is little difference in submission. Reword this.
Discussion
“Little year-to-year change in infection prevalence was noted throughout the 7 years 342 studied, though they represent an increase in past measures. Prevalence estimates in Mas- 343 sachusetts from 2006-2012 were 4.6% for A. phagocytophilum and 1.8% for B. microti [30]. 344 From 2015-2021 however, A. phagocytophilum and B. microti infection rates were higher 345 and remained around 8-9% in adults and 4-6% in nymphs.”
I’m not so sure of this in light of 2B. I think some discussion should be made that your relatively large study conducted using molecular techniques found a large increase in Anaplasma prevalence in ticks.
Most of my work is with Rickettsiaecea and Ehrlichia, so I am not as well versed in this particular subpoint, but at least in the organisms I mention, it is rather well known that PCR is not as reliable as with other pathogens. Some discussion of this point might be warranted in the limitations.
Minor editing of English language required
Author Response
- This is a good manuscript with some important findings. It extends work that has been done with Lyme and does a good job of calling attention to the too-neglected Anaplasma. I think the authors missed an opportunity to highlight the results of FIG2B, but overall this is a very solid manuscript on an increasingly important topic. Please do italicize genus and species names-I count at least 20 instances where this was not done.
The authors appreciate the reviewer raising this issue and apologise with the issues to formatting arising at the time of submission. This issue has been resolved in all occurrences in the manuscript.
- “two slightly rarer diseases caused by I. scapularis”. A minor point here-the disease is not caused by I scapularis, but by anaplasma/babesia. It is transmitted by I scapularis.
The authors appreciate the feedback from the reviewer in this sentence. This has been changed to improve accuracy.
- “Anaplasmosis is caused by the rickettsial bacterium” I never liked calling these rickettsial bacteria. Yes they are members of the Rickettsiales, but I think its important to highlight Anaplasma is part of anaplasmataceae, not rickettsiaeceae. I would prefer you remove the word rickettsial
The authors thank the reviewer for this comment. To amend this section, the use of the word rickettsial has been changed to describe the order and family of Anaplasma. For consistency, the same was applied to Babesia in the preceding sentence.
- “hospitalizing one fourth of symptomatic cases” I would say one fourth of reported cases. Anaplasma surveillance is not very good in general, and it is likely there are asymptomatic cases that go unreported.
The authors thank the reviewer for raising this point on clarity. This section has been revised to satisfy the phrasing recommendation.
- “and confirmed cases had laboratory-confirming diagnostics 120 (such as PCR).” I would like it if you listed out the lab confirming diagnostics here-is immunohistochemistry included? IgG based conversions? Cell culture?
The authors thank the reviewer for pointing this out. The inclusion of more detailed information regarding the human laboratory side of surveillance is important. This information has been included in the revised manuscript, with a new paragraph inserted in section 2.3.
- “Bivariate correlations were used to quantify the relationship between the rates of hu- 147 man-biting I. scapularis” I notice that many times you do not italicize I scapularis (twice in this paragraph alone). Please search the document and properly italicize this.
The authors appreciate the reviewer raising this issue and apologise with the issues to formatting arising at the time of submission. This issue has been resolved in all occurrences in the manuscript.
- “e I. scapularis 164 (19,405), Dermacentor variabilis (3,057), and Amblyomma americanum (461). Of the 165 19,405 human-biting I. scapularis submitted, 77% were adults, 22% were nymphs, and 1% 166 were larvae (Figure 1a).” As above, please italicize genus/species
The authors appreciate the reviewer raising this issue and apologise with the issues to formatting arising at the time of submission. This issue has been resolved in all occurrences in the manuscript.
- “Adults (Mean = 169 20.98 hours) spent an average of 4 more reported hours attached than nymphs and larvae 170 (Meannymphs = 17.28 hours, Meanlarvae = 17.33 hours).” You could easily prove this is significantly different-consider doing so. Depending on data a t test should be sufficient. Please also do so for other findings where you claim differences exist. “Ticks spent an average of 20.3 hours of victim-reported time attached to their human 168 hosts. The distribution was shown to significantly differ by tick life stage. Adults (Mean = 169 20.98 hours) spent an average of 4 more reported hours attached than nymphs and larvae 170 (Meannymphs = 17.28 hours, Meanlarvae = 17.33 hours). The proportion of engorged/replete 171 ticks by life stage, as determined by the laboratory, was around 10% for each life stage. 172 Victims 60 years and older reported much longer attachment (Meanadults = 35 hours, 173 Meannymphs = 27 hours). The laboratory-determined engorgement rates of adult ticks were 174 also twice as high for adults 60 years and older (17%) and for children 10 years and 175 younger (20%). Further considering age, older victims submitted more adults (Mean = 41 176 years) than nymphs (Mean = 34 years) and larvae (Mean = 26 years). Adults were primar- 177 ily found attached to the head, neck and back while nymphs were mainly removed from 178 the lower extremities.” You have all this great data. I would like to see it encapsulated in a table to help the readers.
The authors thank the reviewer for these comments and definitely agree that statistics and table presentation of these data should be included. Statistics for these measures have been added to the revised manuscript, including chi squared and Kruskal wallis tests. A supplemental table has been included for the attachment data.
- “Figure 2B” The prevalence increases here. This is fascinating-given that this was PCR derived, I have some faith that this is a true increase.
The authors thank the reviewer for this comment. The results section has been amended to note that there was an increase in B. microti adult prevalence from 2015 to 2016, but that this increase was not significant.
- “Table 1. County-level nymphal (NIP) and adult (AIP) infection prevalence, cumulative 2015-2021” I think the paper could benefit from a county level map of MA. I am interested in the study but not from the Northeast and I had to stop and go lookup counties and cities to follow this. Inserting a map could help (possibly earlier in the paper).
The authors thank the reviewer for this note and have amended the manuscript to include a county-level map as the new figure 1.
- “The yearly number of towns reporting each disease increased stepwise,” I don’t like the use of stepwise here. I think you might mean consistently?
The authors thank the reviewer for the comment and agree that the use of “stepwise” is not appropriate in this situation. “Consistently” is a more appropriate and accurate descriptor and has been inserted in place of “stepwise.”
- Seasonal disease patterns differed for both diseases, but little inter-annual variation 245 was seen for each. I do not think this is accurate. Consider your figure 2B. There is significant annual variation in prevalence of anaplasmosis. Figure 4 simply shows there is little difference in submission. Reword this. I do not think this is accurate. Consider your figure 2B. There is significant annual variation in prevalence of anaplasmosis. Figure 4 simply shows there is little difference in submission. Reword this.
The authors thank the reviewer for raising this point regarding the clarity of figure 4’s in text description. Section 3.5 has been revised to note that negligible variation in seasonality was observed year to year to be more descriptive than just saying year to year variation. Though figure 1B shows a small change of 1% in the Babesia microti prevalence from 2015 to 2016, this difference is not significant. There are no significant differences in prevalence estimates in any of: nymphal/adult groups for Babesia or Anaplasma throughout the study. This is Better described in section 3.2.
“Little year-to-year change in infection prevalence was noted throughout the 7 years 342 studied, though they represent an increase in past measures. Prevalence estimates in Mas- 343 sachusetts from 2006-2012 were 4.6% for A. phagocytophilum and 1.8% for B. microti [30]. 344 From 2015-2021 however, A. phagocytophilum and B. microti infection rates were higher 345 and remained around 8-9% in adults and 4-6% in nymphs.”
I’m not so sure of this in light of 2B. I think some discussion should be made that your relatively large study conducted using molecular techniques found a large increase in Anaplasma prevalence in ticks. Most of my work is with Rickettsiaecea and Ehrlichia, so I am not as well versed in this particular subpoint, but at least in the organisms I mention, it is rather well known that PCR is not as reliable as with other pathogens. Some discussion of this point might be warranted in the limitations.
The authors thank the reviewer for the comments. PCR is the method for pathogen testing of ticks. It is noted in the discussion that an increase in prevalence was noted from 2006-2012 to the present srudy. These are the only conclusions that we can make with regards to prevalence increase as none were noted during the study period.
Round 2
Reviewer 2 Report
The presentation of the manuscript was imroved according with the inquired mentioned.
No more comments.
Reviewer 3 Report
The authors have responded to all comments made.